# Crop Diversification for Improved Weed Management: A Review

**Gourav Sharma [1], Swati Shrestha [2], Sudip Kunwar [2] and Te-Ming Tseng [3],*** 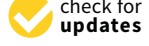

1   School of Plant and Environmental Sciences, Virginia Tech, Blacksburg, VA 24061, USA; gourav1@vt.edu
2   Department of Horticultural Sciences, University of Florida, Gainesville, FL 32611, USA;
    s.shrestha@ufl.edu (S.S.); skunwar@ufl.edu (S.K.)
3   Department of Plant and Soil Sciences, Mississippi State University, Starkville, MS 39762, USA
*   Correspondence: t.tseng@msstate.edu

**Abstract:** Weeds are among the major constraints to any crop production system, reducing productivity and profitability. Herbicides are among the most effective methods to control weeds, and reliance on herbicides for weed control has increased significantly with the advent of herbicide-resistant crops. Unfortunately, over-reliance on herbicides leads to environmental-health issues and herbicide-resistant weeds, causing human health and ecological concerns. Crop diversification can help manage weeds sustainably in major crop production systems. It acts as an organizing principle under which technological innovations and ecological insights can be combined to manage weeds sustainably. Diversified cropping can be defined as the conscious inclusion of functional biodiversity at temporal and/or spatial levels to improve the productivity and stability of ecosystem services. Crop diversification helps to reduce weed density by negatively impacting weed seed germination and weed growth. Additionally, diversified farming systems are more resilient to climate change than monoculture systems and provide better crop yield. However, there are a few challenges to adopting a diversified cropping system, ranging from technology innovations, government policies, farm-level decisions, climate change, and market conditions. In this review, we discuss how crop diversification supports sustainable weed management, the challenges associated with it, and the future of weed management with respect to the diversification concept.

**Keywords:** herbicide resistance; crop diversification; intercropping; crop rotation; cover crops; sustainable; weeds; climate change



## 1. Introduction

Weeds can be defined as any plant that is objectionable or interferes with the activities or welfare of humans [1]. In a crop production system, weeds compete for the same resources as the crops, such as water, nutrients, sunlight, and space, limiting crop productivity [2]. Aggressive weed competition reduces crop yield significantly and adds further cost to crop production owing to their management [3]. Yield loss due to weeds depends on several factors such as density, time of emergence, type of weed, and crop type [4,5]. Globally, up to 40% yield loss because of weeds has been reported [6]. In the USA, yield loss because of weeds has been estimated to exceed eight dollar billion annually [7]. Among US crops, corn and soybean suffer the highest aggregate production loss because of weeds. On average, across 2007–2013, weed interference caused 52 and 50% yield loss in soybean and corn, respectively, in the USA and Canada [8,9]. In Australia and India, annual yield losses due to weeds in grain crops were estimated to be 2.52 and 11 billion USD, respectively [10,11]. China reported a grain loss of approximately 3 million metric tons each year because of weeds [12]. These data indicate that weeds continue to be a major threat in crop production, causing substantial economic and yield loss worldwide [13].

In developing countries, subsistence farming is the primary form of agriculture, and weeds are generally managed through hand-weeding. However, due to increasing urbanization, increased labor costs, and decreasing workforce in agriculture, people are moving

towards using chemicals for controlling weeds. In Southeast Asian countries such as Nepal, Bhutan, Bangladesh, and Thailand, there has been an increase in the haphazard use of herbicides for weed control in subsistence farming systems leading to health and environmental concerns [14,15]. In developed countries such as the USA, China, and Brazil, farmers are engaged in specialized agricultural production systems with the increased use of synthetic fertilizers and herbicides. The top ten consumers of pesticides globally are China, USA, Argentina, Thailand, Brazil, Italy, France, Canada, Japan, and India [16]. In 2014, approximately 2 million tons of chemical pesticides were used in the agricultural sector globally, of which 47.5% constituted herbicides [17]. Over-reliance on herbicides to control weeds and injudicious use of herbicides has led to several issues such as herbicide-resistant weeds, herbicide drift, environmental and health problems, and extinction or population reduction of segetal species [18–20]. Currently, there are approximately 500 unique cases of herbicide-resistant weeds globally [18]. Among the global herbicide-resistant weed cases, most were reported in the USA, followed by Australia, Canada, China, and Brazil [18]. In addition to this, some weeds have developed resistance to multiple modes of action while others have (developed decreased sensitivity to herbicides [21–23]. Both target-site and non-target-site mutations in the herbicide-resistant weeds have been reported [24,25]. These observations indicate over-reliance on herbicides as a non-sustainable measure for weed control. The development of herbicides with novel modes of action is imperative for herbicide resistance management; however, no new mode of action has been developed in the past three decades [26].

Further, there have been increasing reports of crop damage because of herbicide drift in recent years. For instance, 2,4-dichlorophenoxyacetic acid (2,4-D) is one of the most used herbicides to control broadleaf weeds in agriculture; however, they often damage the neighboring 2,4-D sensitive cotton field resulting in the loss of millions of dollars in the USA and Australia [27]. Likewise, severe crop injury has been reported due to the off-target movement of dicamba to the neighboring fields with non-dicamba tolerant crops [28,29]. Further, increasing use of herbicides has led to the accumulation of agricultural contaminants such as arsenic, cadmium, lead, and mercury in soil and water resources [30]. In the USA, a survey of 51 major river basins by the US Geological Survey reported that pesticides were detected in 97% of the samples from streams near agricultural areas [31]. Short and long-term health effects from exposure to agricultural chemicals have also been documented [32,33]. These examples provide evidence that over-dependence on herbicides may result in the increased frequency of herbicide resistance in weeds, water and soil pollution, and herbicide drift. Thus, to mitigate and/or eradicate ecological, environmental, and social externalities associated with intensive use of herbicides, it is imperative to design and promote alternative weed management approaches.

Studies have suggested that increasing crop diversity can subject weeds to a greater number of stresses and reduce reliance on external chemicals for weed/pest control [34,35]. Crop diversification can be defined as the conscious inclusion of functional biodiversity at the temporal and/or spatial levels to improve productivity and stability of ecosystem services [36]. The concept of crop diversification is complex, and a diversified cropping system is more complicated with different crop combinations, unlike monoculture, where extensive farmlands are cultivated with one or two annual crops. Modern agricultural practices have simplified the agricultural systems to enhance the profitability of major crops or livestock. In contrast, a diversified cropping system focuses on creating sustainable, resilient, and socially just global food systems. Some of the examples of a diversified cropping system would be (i) multiple genotypes of the same crop or different crops grown in polyculture [36], (ii) inclusion of legumes in otherwise cereal dominated systems [36] and, (iii) temporal and spatial rotation of crops, including but not limited to cover crops, trap crops, hedgerows, fallow fields, etc. [36]

There are certain set of rules on which crops to choose in a diversified farming system (for, e.g., Liebman and Dyck talks about strategies for crop rotation and intercropping in context of weed management [37]). The consequences of diversification include, but are

not limited to, increased soil nutrient recycling, pest and disease suppression, enhanced water use efficiency, and pollination [38]. Many of the previous studies have advocated the importance of crop diversification in sustainable agricultural production [39,40]. However, knowledge of how different crop diversification techniques impact weed management and the constraints of adopting crop diversification in the modern agricultural context is lacking [41]. This review paper will discuss various crop diversification techniques in terms of weed management and how crop diversification fits in today's era of modern agriculture. The knowledge will provide insights into how crop diversification can be integrated with the present agricultural system for sustainable weed management.

## 2. Crop Diversification Focused on Weed Management

Liebman and Staver [34] noted two general principles for weed management through crop diversification, (a) weeds should be subjected to various stress and mortality factors by using crop sequences containing different species and management practices, and (b) diversification methods should be designed to maximize the capture of light, nutrients, and water by crop, thus reducing the loss by weeds. These principles should be the foundation for any crop diversification methods (e.g., crop rotation, cover cropping, and intercropping). However, the objective of the diversification strategies is not to eliminate all the weeds, instead to control them. Weeds offer various ecosystem services, which sometimes are beneficial to crops and humans [42–47].

### 2.1. Crop Rotation

Crop rotation is the practice of growing a series of crops sequentially over time on the same land, thus providing temporal variability [48]. Crop rotation is a sustainable agricultural practice aimed at achieving high economic output with minimum possible cost [49]. Zhao et al. [50] recently performed a meta-analysis on 45 studies and reported a 20% increment in crop yields due to crop rotation. Moreover, another meta-analysis on 54 studies showed that crop rotation leads to a 49% reduction in weed density [51]. Thus, crop rotation helps to reduce weed pressure and increase crop yield.

Weed species in monoculture tend to adapt to management practices and cause yield reductions (e.g., herbicide resistance, early seed shattering, and crop mimicry). In crop rotations, weeds are subjected to diverse weed control methods (no-tillage/till or diverse herbicides, planting dates, fertilization regime), thus preventing weeds from adapting and surviving [52]. Crop rotation diversifies the selection pressures on weeds by using alternative management tactics, alternating patterns, and timings of soil disturbance, light transmission, and nutrients. Therefore, crop rotation favors the establishment of diverse weed flora rather than dominated by one or few weed species, which in some cases leads to reduced input costs (e.g., herbicide usage) [53,54]. For example, Satorre et al. [55] reported modified weed communities and changed the frequency of some weed populations within the 31 soybean (*Glycine max* L.) fields in Argentina, which rotated with either corn (*Zea mays* L.) or wheat (*Triticum aestivum* L.).

Additionally, crop rotation negatively affects weed abundance, biomass, and density. For instance, corn–soybean-winter wheat rotation in Serbia reduces the number of weed species and biomass compared to continuous corn (CC). Moreover, it increases the yield by 30% compared to CC [56]. Similarly, rice-winter corn rotation suppressed 75% higher weed growth and resulted in 11% higher dry biomass than traditional rice-wheat rotation in India [57]. Unlike herbicides, change in weed community, and density takes time, and it is essential to look at the long-term effect. Simic et al. [56] reported corn, when rotated with winter wheat, causes a 92% reduction of weed biomass. This experiment was performed continuously for 11-year along with corn-soybean (CS) rotation. However, it is not always possible to conduct years-long research trials. In that case, modeling simulations are used, which will consider all growth and environmental parameters. For example, Liebman and Nicholas [58] modeled giant ragweed (*Ambrosia trifida* L.) population dynamics in different crop rotation scenarios. They predicted that to prevent an increase in giant

ragweed density, the minimum control efficacy needed from herbicides or cultivation used in corn and soybean would be 99% in a 2-year corn-soybean, whereas 91% in a 5-year corn-soybean-rye-alfalfa system. Thus, diversified rotations have a higher probability of controlling giant ragweed populations. Several other models can help decide the crop rotations based on the need and geography [59–62].

Soil weed seedbanks preserve propagules for the next generation with the traits such as genetic diversity, long-term seed dormancy, erratic germination, and herbicide resistance/susceptibility [63]. These traits allow weeds to thrive in diverse conditions, including stress from management practices and harsh environments. Crop rotation is one of the various approaches to manage soil seedbanks. For instance, Anderson et al. [64] showed that crop rotations could help reduce the seedbank of annual weeds by balancing the seed production frequency. Anderson [65] reported that weed density could be reduced by utilizing balanced life-cycle intervals in crop rotation design. For example, a 2-year interval rotation of warm-season crops diversifying with different planting dates (e.g., corn to sunflower (*Helianthus* spp. L.)) will increase the diversity of weeds and reduce the viability of weed seeds in the soil seedbank [66,67]. The most advantageous rotation sequences for weed seed management should include four different crops in a series of two warm-season crops followed by cool-season crops [64]. This strategy will help eliminate seed production of warm-season weeds during the two-year cool-season crop and will further decrease during next year's crop [66]. Additionally, crop rotation can help reduce the seedbank density and composition (e.g., Cardina et al. [68] in CC, CS, and corn–oats–hay rotation and Westerman et al. [69] in CS and CS-triticale+ alfalfa-alfalfa).

Crop rotation with a non-host of parasitic plants can help reduce the seedbank of parasitic weeds. The non-host crop is also known as a trap crop, which tricks parasitic seeds to germinate while not causing any crop loss. Oswald and Ransom [70] reported crop rotation as one of the most effective methods to reduce *Striga* infestations in corn and increase corn yields. Samake et al. [71] concluded similar results for *Striga hermonthica* L. in traditional millet and cowpea rotation. Parasitic weeds are genetically diverse and can have differential germination responses to natural stimuli. For example, Hayat et al. [72] showed that the germination of broomrape species (*Orobanche cumana* Wallr and *Phelipanche aegyptiaca* Pers.) responds differently to 2-year crop rotation of sugar beet, pepper, and wheat with sunflower and tomato. *O. cumana* seedbank was reduced after two years, whereas there was no change in the seedbank of *P. aegyptiaca,* thus, making them difficult to manage with the same crop rotation. Crop rotation can also help prevent herbicide resistance by promoting the usage of herbicides with diverse modes of action. Norsworthy et al. [73] listed crop rotation as one of the best management practices to mitigate herbicide resistance. Herbicide-resistant risk is greater where no crop rotation is practiced as compared to fields with crop rotation. For example, modeling simulations have been shown to reduce glyphosate resistance in Palmer amaranth by two-fold if glyphosate-tolerant (GR) cotton is rotated with GR corn [74]. Similarly, blackgrass (*Alopecurus myosuroides* L.) is one of the most essential grass weeds of winter cereal crops in the UK and Europe. Over-reliance on chemical control has led to the widespread development of herbicide resistance in blackgrass. Crop rotation with spring cropping is one of the most effective methods of sustainable blackgrass control in the agricultural system (Table 1) [75]. Balanced crop rotation with spring cropping can reduce the blackgrass population by 78–96% [76]. Reduced blackgrass population in spring cropping is primarily because approximately 80% of the blackgrass germinates in autumn, and thus spring-sown crops are much less infected by blackgrass [75]. Thus, understanding the ecology of weeds and adapting to cultural practices can promote sustainable weed management with less dependency on herbicides.

A survey in Germany indicated that 89% of the farmers use crop rotation to control or prevent herbicide resistance [77]. Similarly, a Canadian farmer's survey in 2015 reported that 80% of the farmers depend on crop rotation to control herbicide resistance in the fields. Canola (*Brassica napus* L.) farmers in Canada use wheat as a rotation crop to gain profit and control ALS enzyme-inhibiting herbicide-resistant weeds [78]. In some cases, crop rotations

can be highly profitable for the farmers. For instance, Goplen et al. [79] showed the average net return of alfalfa-alfalfa-corn (AAC) rotation in herbicide-resistant giant ragweed to be USD 919 ha$^{-1}$ yr$^{-1}$, whereas for CC to be USD 247 ha$^{-1}$ yr$^{-1}$. Moreover, AAC rotations for multiple years led to depleting the herbicide-resistant giant ragweed seedbank [80]. Overall, crop rotation can reduce the risk of herbicide resistance by diversifying weed flora and reducing seedbank.

## 2.2. Intercropping

Intercropping is an integrated weed management practice in which two or more crop species or genotypes are cultivated together and coexisting for a time. It is commonly used in countries with low-input (high-labor) and resource-limited agricultural systems on a small piece of land [81,82]. Intercrops can broadly be divided into three types: (a) relay intercropping (planting a second crop before the first crop is mature), (b) mixed intercropping (simultaneously growing two or more crops), and (c) strip cropping (growing two or more crops simultaneously in strips) Figure 1 [83]. Each type has its benefits, but overall intercropping compared with monocrops provides a similar yield with reduced inputs, pest control (weeds, diseases, and insects), and stable aggregate food yields per unit area [84,85].

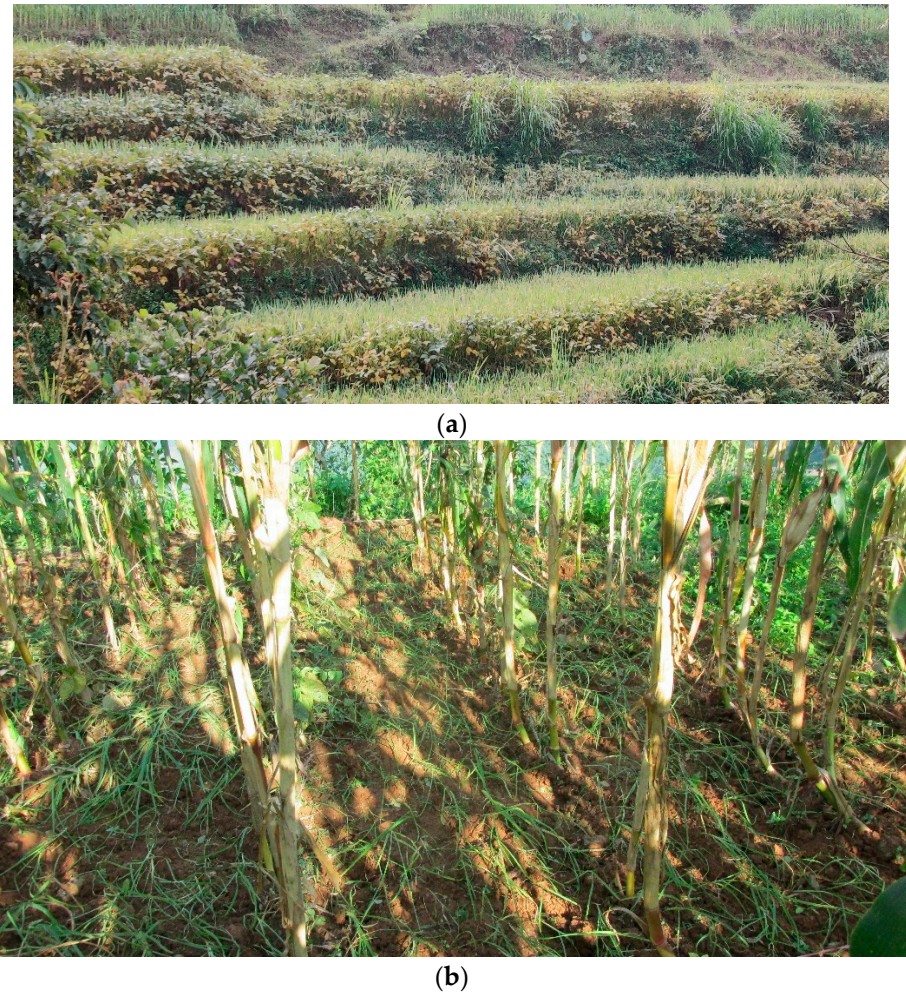

(**a**)

(**b**)

**Figure 1.** (**a**). Intercropping of blackgram and rice (*Oryza sativa* L.) on step farm in Nepal, (**b**). Corn intercropped with ginger (*Zingiber officinale* L.).

In the case of weed management, intercropping reduces weed pressure and allows crops to proliferate. It works based on the principle of resource partitioning among co-

occurring crop species with different resource acquisition strategies, allowing crops to use resources better and leaving less space, water, and nutrients to weeds [86]. Intercropping creates a situation with the increased availability of common limiting factors to the crop [87]. Resource partitioning is more likely to occur when functionally different crops are combined [88,89]. For example, cereals intercropped with legumes can improve nitrogen fixation, better weed control, and high yields [90–92]. Corre-Hellou et al. [93] reported pea (*Pisum sativa* L.)–barley (*Hordeum vulgare* L.) intercropping helped reduce weed biomass by threefold compared to pea monoculture. Similarly, Saucke and Ackermann [94] conclude that pea–alse flax (*Camelina sativa* L.) intercrop had a more significant suppressive effect on weeds than monocrops. Intercropping in drylands can help restore soil fertility and effective weed control. For instance, pearl millet in semi-arid and arid conditions intercropped with short-duration legume crops such as blackgram (*Vigna mungo* L.) and greengram (*Vigna radiata* L.) provides higher yield and reduced weed density and dry weight [95].

Furthermore, the spatial arrangement of the intercrops can significantly affect the yield and weed control. Alternate-row intercropping of soybean with lentil showed the highest yield, whereas the best weed control was reported with sunflower (2018), and buckwheat (2019) (*Fagopyrum esculentum* L.) while maintaining similar soybean yield [96]. Additionally, plant density of crop and intercrop can further affect weed suppression; for example, planting corn at a density of 9 plants/m$^2$ and simultaneously intercropping with cowpea at a density of 30 plants/m$^2$ leads to significant weed suppression when compared with monoculture corn [97].

Apart from the resource partitioning, intercropping may involve allelopathic interactions, which is environmentally friendly and provides economic weed control [98–100]. The mechanism of allelopathy involves both inhibitory and stimulatory relations among neighbor plants, directly affecting growth and development [101]. Among cultivated crops, sorghum species are extensively studied for their allelopathic potential and characterization of allelochemicals associated with weed suppression [102,103]. Several studies have indicated that intercropping sorghum with crops provides better weed control and higher yields than their monocrops [104–106]. Allelopathy is also associated with other field crops such as corn [107,108], oats (*Avena sativa* L.), pearl millet (*Pennisetum glaucum* L.), sesame (*Sesamum indicum* L.) [109], sunflower [104], and soybean [110]. All of the crops mentioned above showed weed suppression when intercropped with other cereals/legumes/oilseeds.

*Striga* spp. are among the notorious weed parasites, which can cause severe damage to crops. Intercropping is one of the effective methods to manage *Striga* for small-scale subsistence farmers. Oswald et al. [111] reported peanut (*Arachis hypogaea* L.), bean (*Phaseolus vulgaris* L.), yellow gram (*Cicer arietinum* L.), Bambara nut (*Vigna subterranean* L.), and soybean, when intercropped with corn, provides high productivity and stable *Striga* control. Furthermore, they indicated shading, higher humidity, and lower temperature under the intercrop canopies were mechanisms that caused a decline in *Striga* numbers. Allelopathic compounds from the Brassicaceae family are proven to be useful for the control of *Orobanche* spp. Fernandez-Aparicio et al. [112] showed that intercropping berseem clover (*Trifolium alexandrinum* L.) with pea, lentil, and chickling pea reduces the infestation of *Orobanche crenata* L. Similarly, witchweed (*Striga hermonthica* L.) haustorial development is inhibited by an aqueous solution eluted from silverleaf (*Desmodium uncinatum* L.). Furthermore, *D. uncinatum*, when intercropped with corn, showed a reduction *in S. hermonthica* infestation [113]. Thus, parasitic weeds can be managed by intercropping, and further investigation of the allelopathic exudates will help develop new bio-herbicides (e.g., *Orobanche*, Cimmino et al.) [114].

### 2.3. Cover Crops

Cover crops are crops planted between the growing seasons to improve soil health, reduce soil erosion, suppress weeds and other pests [115,116]. Cover crops suppress weed growth by multiple mechanisms such as competition for light, space, water, and

nutrients [117]. After the termination of the crop, it forms a mulch layer on the soil surface, which is proven to reduce weed germination, emergence, and establishment [118–120]. Furthermore, cover crops and associated mulch have been shown to release allelochemicals, which further suppress weed growth [121]. A recent meta-analysis on 15 studies covering crop treatment in corn-soybean rotations showed that cover crop helps significantly reduce the weed biomass, but without changing the weed density. Moreover, to achieve a 75% reduction in weed biomass, it requires at least 5 mg ha$^{-1}$ of the cover crop [122].

Cover crops require adequate soil water, moderate temperature, and good seedbed preparation for quick emergence and robust growth [123]. Therefore, the selection of the cover crops depends on field conditions, desirable outcome, and cost. For instance, 759 farmers were surveyed in North Carolina about the perception of cover crops, and 46% of them cited time and labor involved in cover crops as a reason not to adopt it [124]. Moreover, they reported 28.1% of farmers are using cover crops to control weeds. Additionally, Osipitan et al. [125] reported in a meta-analysis of 53 studies that grass cover species provide greater weed suppression compared to broadleaf; fall-sown cover crops provide higher weed suppression than spring-sown cover crops. Moreover, weed suppression increased by increasing the seedling rate of the cover crop from 1 to 3. Thus, cover crop selection and management practices should be meticulously selected while considering cost and labor.

Common cover crops for weed control could be classified in four-group bases taxonomy: cereals, legumes, non-legumes, and brassicaceae plants [126]. One of the cereals, rye (*Cereale secale* L.), is most commonly used as a winter cover crop in soybean and corn as it provides reasonable weed control and yield [127]. Similarly, other commonly used cover crops viz white clover (*Trifolium repens* L.), annual ryegrass (*Lolium multiflorum Lam.* L.), crimson clover (*Trifolium incarnatum* L.), oilseed radish (*Raphanus sativus* L.), sunflower (*Helianthus annuus* L.), field pea, barley and hairy vetch (*Vicia villosa* L.) significantly suppress weeds as compared to monoculture crops [128–131]. Cover crops are planted either alone or in mixtures (to increase crop diversity). Mixture species with different characteristics create a single crop to perform various functions [132]. However, weed management studies have shown that mixture performs better when composed of highly competitive species since biomass is a major predictor of weed suppression [133–136]. Furthermore, Florence and McGuire [137] performed a meta-analysis on 27 studies and found that mixture and monoculture cover crops perform comparably equal in seven metrics (biomass, N, weed, water, biology, yield, and stability). Time and method of cover crop termination can enhance, decrease or have no effect on weed establishment [138]. For example, Wallace et al. [120] reported that delaying cover crop termination can improve weed suppression through a higher accumulation of crop biomass. Cover crops can be terminated climatically, chemically, or mechanically; the appropriate method will depend on the farm management objective [139]. Thus, choice of cover crop, diversity, termination timing, and cover crops strategy can vastly affect weed suppression.

Cover crops are part of an integrated weed management practice, preventing and managing herbicide-resistant weeds (Table 1). For instance, Cholette et al. [140] showed that annual ryegrass alone or in combination with crimson clover leads to reduced density, biomass, and suppression of glyphosate-resistant Canada fleabane (*Erigeron canadensis* L.). However, in some cases, cover crops alone cannot control herbicide-resistant weeds. For example, hairy vetch and crimson clover residues provide early season suppression of glyphosate-resistant Palmer amaranth. However, the integration of herbicide mixtures that incorporate multiple sites of action with cover crops can help mitigate further selection of glyphosate resistance in Palmer amaranth (*Amaranthus palmeri* L.) [141]. Moreover, Bunchek et al. [142] concluded from their study that long-run intensified cover crops could be helpful to manage herbicide resistance since it will help reduce the herbicide selection pressure.

The diversity and size of the weed seedbank strongly influence the success of weed management practice. Cover crops can reduce weed seedbank by preventing propagule production, reducing seedling establishment, early/delay emergence [143]. Long-term

use of the cover crop before cash crops can help to deplete weed seedbank. For instance, Nichols et al. [122] concluded from a 5-year study that long-term use of winter rye in the corn-soybean system has the potential to reduce the size of weed seedbanks compared to winter fallows. Moonen and Barberi et al. [144] reported similar results: after seven years, rye cover crop in corn resulted in a lower seedbank density when compared with the crop residue. However, some studies have noticed no apparent change in weed seedbank [145,146]. Hence, more research efforts are needed to understand the role of cover crops and weed seedbank.

**Table 1.** Crop diversification recommendations to control Palmer Amaranth, Blackgrass, and Rigid Ryegrass.

| Weed Species = Palmer Amaranth (*Amaranthus palmeri*) | | |
|---|---|---|
| **Diversification Strategy \*** | **Details** | **Reference** |
| Crop Rotation in cotton with corn | Crop rotation of cotton and corn reduce the risk of developing glyphosate resistance in Palmer amaranth by ~50% | [74] |
| Cover crops for corn | Hairy vetch and crimson clover provide early season suppression for glyphosate-resistant Palmer Amaranth in corn | [141] |
| Cover crops for cotton | Austrian winterpea (*Lathyrus hirsutus* L.), cereal rye, crimson clover, hairy vetch, oats, (*Avena sativa* L.), rapeseed, (*Brassica napus* L.), and wheat can be used to reduce Palmer amaranth emergence in cotton | [147,148] |
| Cover crop for glyphosate- and dicamba-tolerant (GDT) soybean | Hairy vetch and wheat were effective to control Palmer amaranth. However, cover crop termination and herbicide program should be taken into consideration for maximum yield and highest weed control | [149] |
| Weed Species = Blackgrass (*Alopecurus myosuroides*) | | |
| Cover crop for various crops | Ryegrass as a cover crop can help to reduce the emergence of blackgrass by 17%. It can be used as a cover crop in corn, soybean, and winter-wheat | [150] |
| Crop rotation of winter-annual and spring crops | Five-year rotation with winter wheat, corn, summer barley, winter oilseed rape, and winter wheat reduce blackgrass densities by 50% as compared to winter wheat and winter oilseed rape rotation | [151] |
| Weed Species = Rigid Ryegrass (*Lolium rigidum*) | | |
| Crop rotation | Oaten hay (*Avena sativa*), filed pea, wheat, and barley crop rotation helps to deplete rigid ryegrass seedbank, reduce in-crop weed infestation, and higher profitability | [152] |
| Cover crop in corn | Velvet bean (*Mucuna pruriens* (L.) DC. var. utilis) has allelopathic potential and can help to reduce rigid ryegrass biomass, height, and leaf number | [153] |

\* All diversification strategies were used in combination with a suitable herbicide program.

## 3. Major Constraints to Adoption of Crop Diversification in Modern Agriculture

A large-scale monoculture agriculture system has deeply entrenched across the world ensuing difficulty for any alternative production system such as diversified farming to flourish [154,155]. Reluctance among the commercial farmers in adopting crop diversification could be because of not prioritizing the importance of ecology or the lack of knowledge about diverse farming models and relevant scientific mechanisms governing their several advantages [34,156]. In the absence of proper knowledge and skills, it is evident for the farmers to be more doubtful of the economic success of relatively complex farming systems on a large-scale. In addition, the current agricultural technologies development is mostly centered towards sole-crop farming. For instance, plant breeding tools focusing on improving a few key traits have contributed to increased specialization and reduced genetic diversity [157]. Moreover, too much focus on plant improvement against biotic/ abiotic

stresses is likely to limit the willingness of the farmers to adopt a diversified cropping system to develop crop resilience [40].

Although diversified farming uses agricultural inputs more judiciously and could be cheaper in the long run, small-scale farmers may struggle to establish a diverse agriculture farm as it requires more resources at the beginning. A case study on danish farmers revealed that, with the diversified cropping system, farmers experienced an increase in the types of farm activities, requiring broad knowledge, skills, equipment, manpower, and advisory services to run, and needing more years to return the initial investments [156]. Therefore, farmers with no other alternative income source are unlikely to take risk towards the diversified approach [154]. More naturally grown agricultural products seem to get a high price, and price risk is reduced with the diversification. However, farmers may experience difficulty in transportation and marketing for a small number of diverse products in a country such as the USA, where markets are confined to a limited number of large food-processing, distributing, and retailer firms [158].

Furthermore, agricultural policies are inclined towards industrialized and intensive agriculture, impeding the adoption of crop diversification. Most government incentives and subsidies focus mainly on increasing the production of certain agricultural commodities rather than implementing diversified farming [40,159]. In the United States, 89% of the total amount of subsidies between 1995 to 2005 went to producing five major crops [160]. In different geographies, local farmers following complex agricultural systems have better site-specific knowledge and experiences than anyone [161]. Not having proper policies and channels to disseminate farmers' knowledge to the extension workers and researchers also hinders the adoption of crop diversification. Inadequate legislation provisions to meet the needs of diversified farming are also reported by Aare et al. [156] in Denmark. More restrictions were found regarding the use of certain species or cultivars, thus discouraging diversification. On the other hand, current legislation in most countries allows organic farmers to meet the certification standards without following environmentally sustainable or diversified approaches [36], such as the conservation of weed biodiversity, including the Crop Wild Relatives [162,163] with which the cultivated species can breeding providing more resistant hybrids available for use in agriculture, as the species of the genus *Aegilops* L. progenitor of cultivated wheat [164] or even *Avena* L. sp. pl. of cultivated oats.

## 4. Crop Diversification in the Precision Agriculture Era

As discussed in the previous section, the adoption of crop diversification is hindered to a greater extent because of its labor-intensive and time-consuming constraints, resulting in higher costs. The inadequate research further aggravates the condition to support farmers to enhance their knowledge and skills on diversified farming. However, with the advent of various precision agriculture (PA) tools, many problems associated with conventional methods of field examinations have been mitigated [165,166], which could be helpful in the adoption of crop diversification. Such tools encompass advanced technologies such as Global Positioning Systems (GPS), Geographic Information Systems (GIS), remote sensing, artificial intelligence (AI), machine learning (ML), and simulation modeling [167].

GPS helps to provide the precise location with the help of satellites, and when embodied on other systems, it helps in the site-specific sampling or treatment applications based on location information [167]. On the other hand, GIS is a computer-based hardware and software system used to generate maps based on location data and the attributes of interest [168]. These two tools can be used to generate maps with different kinds of agronomic and other data to provide insights into the spatial and temporal variability of an area so that the plans on diversified cropping can be made accordingly to enhance productivity and profitability.

Remote sensing is a widely used tool in PA, which refers to data collection from a distance and is often based on the reflectance radiations from soil or plant [169]. Such radiations fall under a wide range of wavelengths which are assessed by using different

hand-held sensors or sensors embodied in unmanned aerial vehicles (UAVs) [170,171] and ground robotics [172].

The most widely used sensors and imaging techniques in different agricultural applications are multispectral and hyperspectral cameras, thermal cameras, light detection and ranging sensors (LiDAR), artificial vision sensors, etc. [173]. The raw data obtained with these techniques often need to be processed, and different indices (For example, NDVI) and models are developed using ML, AI, and regression techniques [166,174]. These models can estimate a wide range of agronomic traits and other physio-biological variables. The technique has been used for crop monitoring and high throughput phenotyping [17,167,175] and estimating various agronomic traits such as yield [176,177], canopy dimensions [178], leaf nutrient concentrations [174,179], and biomass [180]. Moreover, they have also been used in disease identification and quantification [181–183], identifying water-related stress [184], scouting weeds and insects [185,186], etc. These techniques can surely help researchers, at least in part, to take data quickly from any complex, diversified farming system conveniently and more economically. Moreover, when farmers employ these tools in their diversified farms, they can monitor their fields more often and identify the required treatments with high accuracy.

Additionally, precision agriculture in a diversified farm will reduce the cost by allowing farmers to use agriculture inputs according to the exact need of the grown crop. For instance, practices such as selective fertilizer applications and selective weed control ensure the optimum application of the treatment, thus preventing their overuse or underuse. Site-specific weed management using autonomous spraying UAVs based on remote-weed mapping has already been developed [187]. Variable-rate fertilizer application techniques based on nutrient maps of the field have also been developed [188]. These techniques would improve the agriculture input efficiency, reduce the losses to the environment, and reduce greenhouse gas emissions [189]. Therefore, besides optimizing the use of agriculture inputs and reducing time and labor requirements, precision agriculture tools also help to attain the principle of sustainable, diversified farming.

Recent progress has shown that PA technologies are more applicable, accurate, and efficient than ever [190]. Even though the current use of PA tools is employed only in highly capitalized larger farms in developed countries, the use of PA in diversified farming could result in sustainable and resilient cropping systems with enhanced productivity. So, future precision agriculture technology development works should focus on diversified farming and, if possible, small-scale farmers by creating more affordable tools.

## 5. Conclusions

Despite high yields and low input cost, the modern monoculture system relies heavily on chemicals for weed control generating human health, environmental and ecological concerns. Herbicide-resistant weeds, increasing health issues associated with agricultural chemicals, water and soil pollution are among the major negative impacts of modern agriculture. New and innovative strategies for sustainable weed management are imperative for sustainable weed management before irreversible damage to humans and the environment. The best strategy for developing a resilient and sustainable production system is adopting diversified farming with ecological weed management options. However, farmers are reluctant to adopt a diversified cropping system because of the requirement of varying skill sets and higher initial investment. Efforts must be taken by both government agencies and the private sectors to promote diversified farming among the commercial and small-scale farmers for developing sustainable farming systems in the future.

**Author Contributions:** Conceptualization, G.S., S.S., S.K. and T.-M.T.; Investigation, G.S., S.S., S.K. and T.-M.T.; Resources, G.S., S.S., S.K. and T.-M.T.; Writing—Original Draft Preparation, G.S., S.S. and S.K.; Writing—Review and Editing, G.S., S.S., S.K. and T.-M.T.; Visualization, G.S., S.S., S.K. and T.-M.T.; Funding Acquisition, T.-M.T. All authors have read and agreed to the published version of the manuscript.

**Funding:** Funding for this project was provided by the Mississippi Agricultural and Forestry Experiment Station, and is based upon work that is supported by the National Institute of Food and Agriculture, U.S. Department of Agriculture, Hatch project under accession number 230100.

**Institutional Review Board Statement:** Not applicable.

**Informed Consent Statement:** Not applicable.

**Data Availability Statement:** Not applicable.

**Conflicts of Interest:** The authors declare no conflict of interest.

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
