# Peer review of "Crop Diversification for Improved Weed Management: A Review"

_agriculture, doi:10.3390/agriculture11050461_

Round 1

Reviewer 1 Report

The review study- “Crop diversification for improved weed management: A Review” is interesting and well structured.  The manuscript is supported with relevant references.  It is addressing the weed management issue through crop diversification.

Author Response

Thank you for your review and your valuable comments.

Reviewer 2 Report

The review is well written and structured. The title of the review fits well into the scope of the Journal. The authors summarized  the start-of-the-art. However, I miss clear recommendations and ideas from the authors to solve actual weed problems with the methods of diverse cropping systems. I would recommend to select a few relevant examples, such as resistant Amaranthus palmeri in the USA, blackgrass or Apera spica-venti in Europe, Avena sterilis/fatua in Australia, ... and give clear recommendations how farmers can solve these problems with different cropping practices. Even in the case of Striga hermonthica, the authors do not give clear recommendations how to change the rotations.

Here are a few speicifc comments:

 L. 99ff: This is not true. There is is clear rule for crop rotations (see e.g. Brinkmann and Roemer). One of the myjor reasons for crop rotatio is self-incompatibility of ALL crop, incluidng maize. The clear evidence for this is the decline effect of monocultures or less diversified crop rotations.

L.194: Moss and Lutman showed in their experiments in the UK that blackgrass was suppressed by replacing winter wheat by spring cereals, because blackgrass is a typical winter-annual weed germinating in fall after a period of warm wheather (to break dormancy). However, you must consider that including spring crop in a pure winter-annual crop rotation provides options for farmers in the fall-to-spring period to control weeds by reapeated stubble tillage, cover cropping or even by using glyphosate on the tillage. These options are not available in a winter cereal, winter oil-seed rape rotation.

Author Response

The authors propose a manuscript titled “Crop diversification for improved weed management: A review”. The article is original, well structured and written. In particular, this study takes into consideration and highlights an interisting topic on weeds. In particular the weeds that are among the major constraints to any crop production system, reducing productivity and profitability. The authors very wisely state that: 1) Unfortunately, over-reliance on herbicides leads to environmental-health issues and herbicide-resistant weeds, causing human-health and ecological concerns; 2) Crop diversification can help manage weeds sustainably in major crop production systems. 3) manage weeds sustainably. I personally completely agree with the authors. I analyzed the text and gave some few suggestions in order to publishing the manuscript. The concepts are expressed in the correct way on a very interesting and actual topic.

Introduction:

Please add a references in following periods.

Line 31. The authors declare: “Weeds can be defined as any plant growing where they are undesired”.

Below is the reference added in the manuscript

  • [1] Weed Science Society of America [WSSA] (2021) WSSA Glossary. https://wssa.net/wssa/wssa-glossary/

Lines 33-35. “Aggressive weed competition reduces crop yield significantly and adds further cost to crop production owing to their management.”

Below is the reference added in the manuscript

  • [4] Chauhan, B. S. (2020). Grand challenges in weed management. Frontiers in Agronomy1, 3.

Lines 43-45. “These data indicate that weeds continue to be a major threat in crop production, causing substantial economic and yield loss worldwide”.

Below is the reference added in the manuscript

  • [13] Ramesh, K., Matloob, A., Aslam, F., Florentine, S. K., & Chauhan, B. S. (2017). Weeds in a changing climate: vulnerabilities, consequences, and implications for future weed management. Frontiers in plant science8, 95.

Lines 57-59. Please add a reference for this statement and complete the concept in the suggested way: “Over-reliance on herbicides to control weeds and injudicious use of herbicides has led to several issues such as herbicide- resistant weeds, herbicide drift, and environmental and health problems as well as extinction or pupulation reduction of segetal species [15,16, Perrino and Calabrese 2018]

Reference to be added

Below is the reference added in the manuscript

  • Perrino, E.V.; Calabrese, G. Endangered segetal species in southern Italy: distribution, conservation status, trends, actions and ethnobotanical notes.  Crop Evol201865(8), 2107–2134. Doi: 10.1007/s10722-018-0678-6

Line 99-110. Please add a references for crucial concepts that I suppose there are not a discovers from authors.

Below are references added in the manuscript

  • [41] Meynard, J. M., Messéan, A., Charlier, A., Charrier, F., Farès, M., Le Bail, M., ... & Savini, I. (2013). Crop diversification: obstacles and levers. Study of farms and supply chains. Synopsis of the study report. INRA, Paris.
  • [39] Bommarco, R., Kleijn, D., & Potts, S. G. (2013). Ecological intensification: harnessing ecosystem services for food security. Trends in ecology & evolution28(4), 230-238.
  • [40] Lin, B. B. (2011). Resilience in agriculture through crop diversification: adaptive management for environmental change. BioScience61(3), 183-193.

Please pay attention for this concept and spend two words about it:

Weeds in small percentages contain essential oils useful for human health, and guarantee the development of antibodies. So a yield of the harvest of 100% of wheat or other cereal is not in line with the quality of productivity (see Perrino and Calabrese 2018 and other works)

2.1. Crop Rotation

Line 184. Some weed species could also be eaten directly by man as perhaps Orobanche cumana(see other species of Orobanche), which would be a resource for humans.

2.2. Intercropping

Concept that we must considering. Weeds must never be completely eliminated but only controlled, among other things they often attract pathogens preventing them from going on the cultivated species

All the above-mentioned points were addressed by adding suitable references. We are suggesting that weeds provide beneficiary ecosystem services to humans and crops. Below are references added in the manuscript [42-48]

  • Blaix, C., Moonen, A. C., Dostatny, D. F., Izquierdo, J., Le Corff, J., Morrison, J., ... & Westerman, P. R. (2018). Quantification of regulating ecosystem services provided by weeds in annual cropping systems using a systematic map approach. Weed research58(3), 151-164.
  • Capinera, J. L. (2005). Relationships between insect pests and weeds: an evolutionary perspective. Weed science, 53(6), 892-901.
  • Bretagnolle, V., & Gaba, S. (2015). Weeds for bees? A review. Agronomy for Sustainable Development35(3), 891-909.
  • Smith, B. M., Aebischer, N. J., Ewald, J., Moreby, S., Potter, C., & Holland, J. M. (2020). The potential of arable weeds to reverse invertebrate declines and associated ecosystem services in cereal crops. Frontiers in sustainable food systems3, 118.
  • Mouritsen, O. G. (2017). Those tasty weeds. Journal of Applied Phycology29(5), 2159-2164.
  • Amato-Lourenco, L. F., Ranieri, G. R., de Oliveira Souza, V. C., Junior, F. B., Saldiva, P. H. N., & Mauad, T. (2020). Edible weeds: Are urban environments fit for foraging? Science of the Total Environment698, 133967.

Line 255. Pennisetum glaucum L. (not L but L., L.=Linneus). In this case the authors  use the scientific name in the complete form with the author (in this case L.). Please check whole document and decide if use or not the complete scientifc name. I suggest to use the complete name when the species is cited for the first time.

We have corrected the manuscript as suggested by the reviewer.

2.3. Cover crops

Line 299. Please see my previous comment. (Cereale secale L.). Also remember that L. not in italic, but only Cereal secale. Please reported correctly the scientific name.

  1. Major constraints to adoption of crop diversification in modern agriculture

Lines 379-381. Please complete the period with crucial concept in the suggested way: “On the other hand, current legislation in most countries allows organic farmers to meet the certification standards without following environmentally sustainable or diversified approaches [32], such as the conservation of weed biodiversity, including the Crop Wild Relatives (Maxted and Kell 2009, Hajjar and Hodgkin 2007) with which the cultivated species can breeding providing more resistant hybrids available for use in agriculture, as the species of the genus Aegilops L. progenitor of cultivated wheat (Perrino et al. 2014), or even Avena L. sp. pl. of cultivated oats (Avena sativa L.).

Reference to be added

Below are references added in the manuscript based on the reviewer comments [165-167]

  • Maxted, N.; Kell, S. Establishment of a global network for the in situ conservation of crop wild relatives: status and needs, Rome, Italy: FAO Commission on Genetic Resources for Food and Agriculture. 2009.
  • Hajjar, R.; Hodgkin, T. The use of wild relatives in crop improvement : A survey of developments over the last 20 years. Euphytica2007, 156, 1–13.
  • Perrino, E.V.; Wagensommer, R.P.; Medagli, P. The genus Aegilops(Poaceae) in Italy: taxonomy, geographical distribution, ecology, vulnerability and conservation. Systematics and Biodiversity 2014, 12(3), 331-349.

Reviewer 3 Report

The authors propose a manuscript titled “Crop diversification for improved weed management: A review”. The article is original, well structured and written. In particular, this study takes into consideration and highlights an interisting topic on weeds. In particular the weeds that are among the major constraints to any crop production system, reducing productivity and profitability. The authors very wisely state that: 1) Unfortunately, over-reliance on herbicides leads to environmental-health issues and herbicide-resistant weeds, causing human-health and ecological concerns; 2) Crop diversification can help manage weeds sustainably in major crop production systems. 3) manage weeds sustainably. I personally completely agree with the authors. I analyzed the text and gave some few suggestions in order to publishing the manuscript. The concepts are expressed in the correct way on a very interesting and actual topic.

Introduction:

Please add a references in following periods.

Line 31. The authors declare: “Weeds can be defined as any plant growing where they are undesired”.

Lines 33-35. “Aggressive weed competition reduces crop yield significantly and adds further cost to crop production owing to their management.”

Lines 43-45. “These data indicate that weeds continue to be a major threat in crop production, causing substantial economic and yield loss worldwide”.

Lines 57-59. Please add a reference for this statement and complete the concept in the suggested way: “Over-reliance on herbicides to control weeds and injudicious use of herbicides has led to several issues such as herbicide- resistant weeds, herbicide drift, and environmental and health problems as well as extinction or pupulation reduction of segetal species [15,16, Perrino and Calabrese 2018]

Reference to be added

  • Perrino, E.V.; Calabrese, G. Endangered segetal species in southern Italy: distribution, conservation status, trends, actions and ethnobotanical notes. Resour. Crop Evol. 2018, 65 (8), 2107–2134. Doi: 10.1007/s10722-018-0678-6

Line 99-110. Please add a references for crucial concepts that I suppose there are not a discovers from authors.

Please pay attention for this concept and spend two words about it:

Weeds in small percentages contain essential oils useful for human health, and guarantee the development of antibodies. So a yield of the harvest of 100% of wheat or other cereal is not in line with the quality of productivity (see Perrino and Calabrese 2018 and other works)

2.1. Crop Rotation

Line 184. Some weed species could also be eaten directly by man as perhaps Orobanche cumana(see other species of Orobanche), which would be a resource for humans.

2.2. Intercropping

Concept that we must considering. Weeds must never be completely eliminated but only controlled, among other things they often attract pathogens preventing them from going on the cultivated species

Line 255. Pennisetum glaucum L. (not L but L., L.=Linneus). In this case the authors  use the scientific name in the complete form with the author (in this case L.). Please check whole document and decide if use or not the complete scientifc name. I suggest to use the complete name when the species is cited for the first time.

2.3. Cover crops

Line 299. Please see my previous comment. (Cereale secale L.). Also remember that L. not in italic, but only Cereal secale. Please reported correctly the scientific name.

3. Major constraints to adoption of crop diversification in modern agriculture

Lines 379-381. Please complete the period with crucial concept in the suggested way: “On the other hand, current legislation in most countries allows organic farmers to meet the certification standards without following environmentally sustainable or diversified approaches [32], such as the conservation of weed biodiversity, including the Crop Wild Relatives (Maxted and Kell 2009, Hajjar and Hodgkin 2007) with which the cultivated species can breeding providing more resistant hybrids available for use in agriculture, as the species of the genus Aegilops L. progenitor of cultivated wheat (Perrino et al. 2014), or even Avena L. sp. pl. of cultivated oats (Avena sativa L.).

Reference to be added

  • Maxted, N.; Kell, S. Establishment of a global network for the in situ conservation of crop wild relatives: status and needs, Rome, Italy: FAO Commission on Genetic Resources for Food and Agriculture. 2009.
  • Hajjar, R.; Hodgkin, T. The use of wild relatives in crop improvement : A survey of developments over the last 20 years. Euphytica 2007, 156, 1–13.
  • Perrino, E.V.; Wagensommer, R.P.; Medagli, P. The genus Aegilops (Poaceae) in Italy: taxonomy, geographical distribution, ecology, vulnerability and conservation. Systematics and Biodiversity 2014, 12(3), 331-349.

Author Response

Reviewer 2

The review is well written and structured. The title of the review fits well into the scope of the Journal. The authors summarized  the start-of-the-art. However, I miss clear recommendations and ideas from the authors to solve actual weed problems with the methods of diverse cropping systems. I would recommend to select a few relevant examples, such as resistant Amaranthus palmeri in the USA, blackgrass or Apera spica-venti in Europe, Avena sterilis/fatua in Australia, ... and give clear recommendations how farmers can solve these problems with different cropping practices. Even in the case of Striga hermonthica, the authors do not give clear recommendations how to change the rotations.

We have added a table presenting farmer recommendations for different crop diversification strategies (Table 1)

Here are a few speicifc comments:

  1. 99ff: This is not true. There is is clear rule for crop rotations (see e.g. Brinkmann and Roemer). One of the myjor reasons for crop rotatio is self-incompatibility of ALL crop, incluidng maize. The clear evidence for this is the decline effect of monocultures or less diversified crop rotations.

Added a citation [37]

L.194: Moss and Lutman showed in their experiments in the UK that blackgrass was suppressed by replacing winter wheat by spring cereals, because blackgrass is a typical winter-annual weed germinating in fall after a period of warm wheather (to break dormancy). However, you must consider that including spring crop in a pure winter-annual crop rotation provides options for farmers in the fall-to-spring period to control weeds by reapeated stubble tillage, cover cropping or even by using glyphosate on the tillage. These options are not available in a winter cereal, winter oil-seed rape rotation.

This section was revised to incorporate reviewer suggestions. Additionally, a citation of Moss and Lutman has been added [75,76]

Reviewer 3

The authors propose a manuscript titled “Crop diversification for improved weed management: A review”. The article is original, well structured and written. In particular, this study takes into consideration and highlights an interisting topic on weeds. In particular the weeds that are among the major constraints to any crop production system, reducing productivity and profitability. The authors very wisely state that: 1) Unfortunately, over-reliance on herbicides leads to environmental-health issues and herbicide-resistant weeds, causing human-health and ecological concerns; 2) Crop diversification can help manage weeds sustainably in major crop production systems. 3) manage weeds sustainably. I personally completely agree with the authors. I analyzed the text and gave some few suggestions in order to publishing the manuscript. The concepts are expressed in the correct way on a very interesting and actual topic.

Introduction:

Please add a references in following periods.

Line 31. The authors declare: “Weeds can be defined as any plant growing where they are undesired”.

Below is the reference added in the manuscript

  • [1] Weed Science Society of America [WSSA] (2021) WSSA Glossary. https://wssa.net/wssa/wssa-glossary/

Lines 33-35. “Aggressive weed competition reduces crop yield significantly and adds further cost to crop production owing to their management.”

Below is the reference added in the manuscript

  • [4] Chauhan, B. S. (2020). Grand challenges in weed management. Frontiers in Agronomy1, 3.

Lines 43-45. “These data indicate that weeds continue to be a major threat in crop production, causing substantial economic and yield loss worldwide”.

Below is the reference added in the manuscript

  • [13] Ramesh, K., Matloob, A., Aslam, F., Florentine, S. K., & Chauhan, B. S. (2017). Weeds in a changing climate: vulnerabilities, consequences, and implications for future weed management. Frontiers in plant science8, 95.

Lines 57-59. Please add a reference for this statement and complete the concept in the suggested way: “Over-reliance on herbicides to control weeds and injudicious use of herbicides has led to several issues such as herbicide- resistant weeds, herbicide drift, and environmental and health problems as well as extinction or pupulation reduction of segetal species [15,16, Perrino and Calabrese 2018]

Reference to be added

Below is the reference added in the manuscript

  • Perrino, E.V.; Calabrese, G. Endangered segetal species in southern Italy: distribution, conservation status, trends, actions and ethnobotanical notes.  Crop Evol201865(8), 2107–2134. Doi: 10.1007/s10722-018-0678-6

Line 99-110. Please add a references for crucial concepts that I suppose there are not a discovers from authors.

Below are references added in the manuscript

  • [41] Meynard, J. M., Messéan, A., Charlier, A., Charrier, F., Farès, M., Le Bail, M., ... & Savini, I. (2013). Crop diversification: obstacles and levers. Study of farms and supply chains. Synopsis of the study report. INRA, Paris.
  • [39] Bommarco, R., Kleijn, D., & Potts, S. G. (2013). Ecological intensification: harnessing ecosystem services for food security. Trends in ecology & evolution28(4), 230-238.
  • [40] Lin, B. B. (2011). Resilience in agriculture through crop diversification: adaptive management for environmental change. BioScience61(3), 183-193.

Please pay attention for this concept and spend two words about it:

Weeds in small percentages contain essential oils useful for human health, and guarantee the development of antibodies. So a yield of the harvest of 100% of wheat or other cereal is not in line with the quality of productivity (see Perrino and Calabrese 2018 and other works)

2.1. Crop Rotation

Line 184. Some weed species could also be eaten directly by man as perhaps Orobanche cumana(see other species of Orobanche), which would be a resource for humans.

2.2. Intercropping

Concept that we must considering. Weeds must never be completely eliminated but only controlled, among other things they often attract pathogens preventing them from going on the cultivated species

All the above-mentioned points were addressed by adding suitable references. We are suggesting that weeds provide beneficiary ecosystem services to humans and crops. Below are references added in the manuscript [42-48]

  • Blaix, C., Moonen, A. C., Dostatny, D. F., Izquierdo, J., Le Corff, J., Morrison, J., ... & Westerman, P. R. (2018). Quantification of regulating ecosystem services provided by weeds in annual cropping systems using a systematic map approach. Weed research58(3), 151-164.
  • Capinera, J. L. (2005). Relationships between insect pests and weeds: an evolutionary perspective. Weed science, 53(6), 892-901.
  • Bretagnolle, V., & Gaba, S. (2015). Weeds for bees? A review. Agronomy for Sustainable Development35(3), 891-909.
  • Smith, B. M., Aebischer, N. J., Ewald, J., Moreby, S., Potter, C., & Holland, J. M. (2020). The potential of arable weeds to reverse invertebrate declines and associated ecosystem services in cereal crops. Frontiers in sustainable food systems3, 118.
  • Mouritsen, O. G. (2017). Those tasty weeds. Journal of Applied Phycology29(5), 2159-2164.
  • Amato-Lourenco, L. F., Ranieri, G. R., de Oliveira Souza, V. C., Junior, F. B., Saldiva, P. H. N., & Mauad, T. (2020). Edible weeds: Are urban environments fit for foraging? Science of the Total Environment698, 133967.

Line 255. Pennisetum glaucum L. (not L but L., L.=Linneus). In this case the authors  use the scientific name in the complete form with the author (in this case L.). Please check whole document and decide if use or not the complete scientifc name. I suggest to use the complete name when the species is cited for the first time.

We have corrected the manuscript as suggested by the reviewer.

2.3. Cover crops

Line 299. Please see my previous comment. (Cereale secale L.). Also remember that L. not in italic, but only Cereal secale. Please reported correctly the scientific name.

  1. Major constraints to adoption of crop diversification in modern agriculture

Lines 379-381. Please complete the period with crucial concept in the suggested way: “On the other hand, current legislation in most countries allows organic farmers to meet the certification standards without following environmentally sustainable or diversified approaches [32], such as the conservation of weed biodiversity, including the Crop Wild Relatives (Maxted and Kell 2009, Hajjar and Hodgkin 2007) with which the cultivated species can breeding providing more resistant hybrids available for use in agriculture, as the species of the genus Aegilops L. progenitor of cultivated wheat (Perrino et al. 2014), or even Avena L. sp. pl. of cultivated oats (Avena sativa L.).

Reference to be added

Below are references added in the manuscript based on the reviewer comments [165-167]

  • Maxted, N.; Kell, S. Establishment of a global network for the in situ conservation of crop wild relatives: status and needs, Rome, Italy: FAO Commission on Genetic Resources for Food and Agriculture. 2009.
  • Hajjar, R.; Hodgkin, T. The use of wild relatives in crop improvement : A survey of developments over the last 20 years. Euphytica2007, 156, 1–13.
  • Perrino, E.V.; Wagensommer, R.P.; Medagli, P. The genus Aegilops(Poaceae) in Italy: taxonomy, geographical distribution, ecology, vulnerability and conservation. Systematics and Biodiversity 2014, 12(3), 331-349.
